# Small Batch Size Training for Language Models: When Vanilla SGD Works, and Why Gradient Accumulation Is Wasteful

**Martin Marek**
New York University
martin.m@nyu.edu

**Sanae Lotfi**
New York University

**Aditya Somasundaram**
Columbia University

**Andrew Gordon Wilson**
New York University

**Micah Goldblum**
Columbia University

## Abstract

Conventional wisdom dictates that small batch sizes make language model pretraining and fine-tuning unstable, motivating gradient accumulation, which trades off the number of optimizer steps for a proportional increase in batch size. While it is common to decrease the learning rate for smaller batch sizes, other hyperparameters are often held fixed. In this work, we revisit small batch sizes all the way down to batch size one, and we propose a rule for scaling Adam hyperparameters to small batch sizes. In particular, rather than holding the decay rate of the second moment fixed across batch sizes, we propose to hold its half-life fixed in terms of tokens. We find that small batch sizes (1) train stably, (2) are consistently more robust to hyperparameter choices, (3) achieve equal or better per-FLOP performance than larger batch sizes, and (4) notably enable stable language model training with vanilla SGD, even without momentum, despite storing no optimizer state. Building on these results, we provide practical recommendations for selecting a batch size and setting optimizer hyperparameters. We further recommend against gradient accumulation unless training on multiple devices with multiple model replicas. Finally, we show that a small batch size combined with an optimizer with a small state size can provide the performance benefits of full fine-tuning while maintaining a similar memory footprint to LoRA.

## 1   Introduction

Large batch sizes are widely believed to improve the stability of language model training [1, 2, 3, 4]. As a consequence, sophisticated optimizers that perform well in large-batch training are increasingly standard practice [5, 6, 7, 8, 9, 10]. In fact, it is common to simulate batch sizes even larger than the maximum batch size that fits into device memory through gradient accumulation [4, 3, 11, 12].

In small batch size pretraining experiments, one may observe loss spikes and heavy instability [13, 14, 15, 16]. While it is common to decrease the learning rate for smaller batch sizes, other hyperparameters, such as the decay rates for the first and second moments in Adam ($\beta_1$ and $\beta_2$), are often held fixed across batch sizes. We show that if instead of holding $\beta_2$ fixed, we hold the *half-life* of $\beta_2$ fixed (measured in number of tokens), stable training is possible all the way down to batch size one, as illustrated in Figure 1a.

When scaling hyperparameters in this way, we find that small batch sizes can confer computational advantages and greater robustness. In the small batch regime, we observe that the speed of convergence is less sensitive to optimizer hyperparameters like the learning rate, and momentum becomes

39th Conference on Neural Information Processing Systems (NeurIPS 2025).

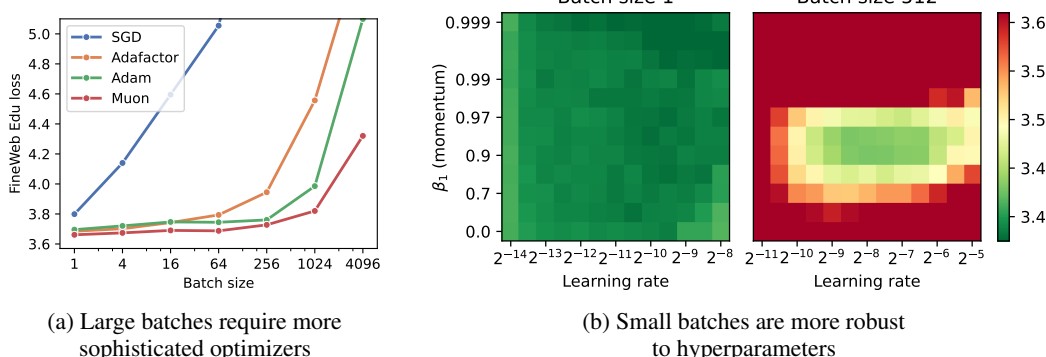

(a) Large batches require more
sophisticated optimizers

(b) Small batches are more robust
to hyperparameters

Figure 1: **Small batch sizes are more robust to optimizer design and hyperparameter values. (a)** Loss achieved by a transformer decoder-only language model with 30M active parameters trained on 600M tokens of FineWeb-Edu data using SGD, Adafactor, Adam, and Muon. At small batch sizes, all optimizers achieve similar loss; at large batch sizes, the gap between optimizers grows. **(b)** Loss achieved by GPT-2 (124M) trained on 2.5B tokens of FineWeb using AdamW when tuning the learning rate and $\beta_1$ hyperparameters for batch sizes 1 and 512. While both batch sizes achieve a similar lowest loss, the smaller batch size is much more robust to the hyperparameter values, reducing the need for hyperparameter tuning. For batch size 512, we use the default hyperparameters from Brown et al. [17] and the nanoGPT public repository [18]. For batch size 1, we turn off weight decay and rescale $\beta_2$ to preserve the token half-life, resulting in $\beta_2 = 0.9999$.

less necessary, as we find in Figure 1b. In contrast, large batch sizes require large learning rates to compensate for taking fewer steps at a given compute budget. Large step sizes in turn require that the optimizer make predictions about the loss surface far away from the current iterate. We find that predicting such far-away parameter updates requires a sophisticated and carefully tuned optimizer. For very small batch sizes, we find that even vanilla stochastic gradient descent (SGD), with no momentum or weight decay, becomes competitive for training language models, in contrast to findings from recent work [19]. The ability to perform stable training without momentum can confer significant memory advantages, since we do not need to store the corresponding optimizer state, which can have a significant memory footprint. Based on our observations, we also revisit Adafactor, a memory-efficient variant of Adam, and show that it can be a compelling alternative to Adam in the small batch regime, enabling training of larger models in a memory-constrained setting.

These findings could have a significant bearing on practice. Rather than perform gradient accumulation or use the largest batch size that fits into device memory, it could be preferable to use smaller batch sizes that enable simpler procedures, computational and memory advantages, and require less tuning. Our findings suggest a best practice of using the smallest batch size that maximizes model throughput. We find that these prescriptions hold in both LLM pretraining and fine-tuning.

It is common in convex optimization to use higher batch sizes as the loss converges to a minimum [1, 20]. Intuitively, the higher the gradient to gradient noise magnitude ratio, the smaller the batch size we should take. For example, if the gradient noise magnitude is close to zero, then we obtain roughly the same gradient estimate with a small batch size as a large batch size but pay far fewer floating-point operations (FLOPs), so small batch sizes are efficient. Smaller batch sizes allow us to take more steps for a fixed FLOP budget. As we converge to a minimum, that same ratio instead often tends to zero since the gradient norm vanishes while the gradient noise may not, in which case we may want a larger batch size. We hypothesize that language model training lives in the far-from-convergence regime where small batch sizes are efficient. Following compute-optimal scaling laws, we would not train language models into the convergence regime since we could achieve better performance by instead training a larger model on less data [21, 22].

The paper is structured as follows: in Sections 2 and 3, we cover the background and related work for language model training. In Section 4, we demonstrate that small batch sizes not only perform on par with larger batch sizes for several optimizers that we carefully tune, but they also exhibit more robustness to optimizer and hyperparameter choices. We also show that the gap between vanilla SGD and more sophisticated optimizers shrinks in the small batch regime. Moreover, we derive scaling heuristics for Adam's hyperparameters and demonstrate that they yield improved performance when transferred to new, larger-scale settings. Finally, we apply these results to memory-efficient

fine-tuning using a small batch size and Adafactor, which achieves the best trade-off between memory footprint and performance. Motivated by our findings, we provide practical prescriptions for training language models in Section 5. Namely, we recommend using the smallest batch size that maximizes model throughput and advise against gradient accumulation for most practitioners.

## 2 Background: Optimization for Language Models

In this section, we briefly review relevant optimization algorithms for language model training and the role of their hyperparameters.

**Stochastic gradient descent (SGD)** [23, 24]. SGD updates model parameters by computing gradients on mini-batches of data as follows: $\theta_{t+1} = \theta_t - \eta g_t$, where $\theta_t$ are the model parameters at step $t$, $\eta$ is the learning rate, and $g_t = \nabla_\theta \mathcal{L}(\theta_t)$ is the gradient of the loss function with respect to $\theta_t$ averaged over $B$ samples. We refer to $B$ as the batch size and consider the extreme case of $B = 1$ in our experiments.

We show that although SGD is extremely simple, it can perform competitively when correctly tuned at small batch sizes. Momentum is often added to accumulate a moving average of past gradients, smoothing updates, and accelerating convergence. However, we focus on vanilla SGD in our experiments to make the point that, with a small batch size, even momentum is unnecessary.

**Adam** [25]. Adam is an adaptive optimizer that maintains an exponential moving average (EMA) of the gradient and squared gradient, referred to as the first and second moments, and denoted as $m_t$ and $v_t$. The timescales of these moving averages are controlled by decay rates $\beta_1$ and $\beta_2$:

$$m_t = \beta_1 m_{t-1} + (1 - \beta_1)g_t$$
$$v_t = \beta_2 v_{t-1} + (1 - \beta_2)g_t^2. \tag{1}$$

The model parameters are then updated as follows: $\theta_{t+1} = \theta_t - \eta \frac{m_t}{\sqrt{v_t} + \epsilon}$, where $\epsilon$ is a small constant to prevent division by zero.

Higher values of $\beta_1$ and $\beta_2$ place more weight on past gradients, meaning that the moments are averaged over longer timescales and evolve slowly. When training language models, values like $\beta_1 = 0.9$ and $\beta_2 \in [0.95, 0.98]$ are often used in practice [17, 26]. While most researchers and practitioners do not scale $\beta_1$ and $\beta_2$ with the batch size, we show in our work that scaling $\beta_2$ in particular is crucial for achieving good performance with Adam at a small batch size.

**Moment half-life**. We found it helpful across many of our experiments to measure the typical timescales (measured in number of tokens) that the first and second moments in Adam are averaged over, instead of directly working with $\beta_1$ and $\beta_2$.

At each Adam update step [eq. (1)], the contribution of previous gradients to the first moment gets reduced by a factor of $\beta_1$ and the contribution to the second moment gets reduced by a factor of $\beta_2$. Hence, by definition, there exists a certain number of steps $n$, after which the contribution of any mini-batch gradient is halved: $\beta^n = \frac{1}{2}$. Since every update step corresponds to observing $(B \cdot T)$ tokens, where $B$ is the batch size and $T$ is the sequence length, we can convert the number of optimizer steps into a number of observed tokens. As such, we can express the number of tokens it takes to decay a mini-batch gradient by a factor of $\frac{1}{2}$ as the *half-life* of the decay rate, denoted by $t_{1/2}$, where $\beta^{\frac{t_{1/2}}{BT}} = \frac{1}{2}$. The half-life provides a measure of the "typical" timescale that gradients are averaged over. For

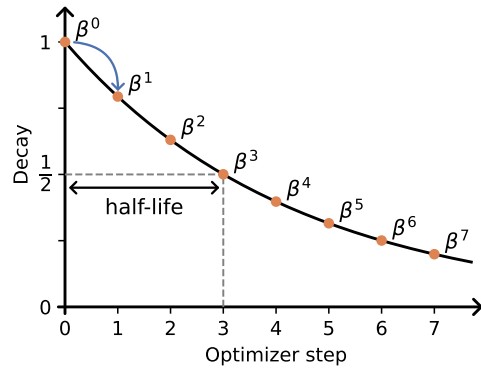

Figure 2: **Moment half-life**. The first and second moments in the Adam optimizer are exponential moving averages of past mini-batch gradients. At each step, the contribution of past mini-batch gradients decays by a factor of $\beta$. After a given number of optimizer steps (or equivalently, after a given number of training tokens), the contribution of any mini-batch gradient will decay by a factor of $\frac{1}{2}$. We call this number of tokens the decay *half-life*.

example, if we use a half-life of $t_{1/2} = 10M$ tokens, this means that the second moment is averaged over *roughly* 10M tokens. Throughout this paper, we will refer to the half-life of the first moment as $t_1$ and the half-life of the second moment as $t_2$, analogously to the more conventional parameterization $\beta_1$ and $\beta_2$.

**Adafactor** [27]. The second moment estimate $v_t$ used in Adam requires storing a number of floating point values equal to the parameter count of the model itself. Instead, Adafactor only stores the per-row and per-column sums of the moving average for the neural network weight matrices and estimates the per-parameter second moment using these sums. Adafactor also does not store a first moment estimate at all. Therefore, for a weight matrix with shape $d_1 \times d_2$, the memory requirements for the moving averages are reduced from $2 \times d_1 \times d_2$ for Adam to $d_1 + d_2$ for Adafactor. We demonstrate in Section 4.5 that thanks to this sublinear memory cost, Adafactor achieves the best performance–memory trade-off.

**Muon** [9]. Another optimizer that was recently demonstrated to be competitive for language model training is MomentUm Orthogonalized by Newton-Schulz (Muon). Muon is specifically designed for $\geq 2$-dimensional parameter tensors, such as weight matrices in linear layers, where the Newton-Schulz iterative method is applied to the first moment estimate $m_t$ before using it to update the parameters. Muon is often used in conjunction with other optimizers like Adam. In our experiments using Muon, we only apply it to hidden layers and use Adam for input embedding and final layers.

**Gradient accumulation**. When hardware memory constraints prevent us from using large batch sizes directly, we can simulate a large batch size by summing gradients across multiple successive micro-batches and only periodically performing an optimizer step. Under gradient accumulation, the effective batch size is equal to the micro-batch size multiplied by the number of gradient accumulation steps. While gradient accumulation allows for larger effective batch sizes, it still increases memory usage compared to using a small batch size, because it requires storing the accumulated gradients.

## 3   Related Work

Batch size has long been studied in both theory and practice: its impact on the optimization path [28, 29, 30, 31], its interaction with different optimizers [32, 33, 34, 35], and its effect on the optimal learning rates and momentum hyperparameters [36, 37, 38, 39, 34, 40, 41, 42, 41, 43, 44]. Prior work has also explored related themes, such as the flatness argument favoring SGD [45, 46, 47, 48], SGD's implicit biases [49, 50, 51, 52], and the comparison between stochastic and full-batch optimization [53, 54, 55, 56]. However, modern language model training is neither convex nor characterized by multiple epoch training like vision settings. In fact, we often train for a limited token budget and stop far before convergence, since otherwise scaling laws suggest that we could train a much better language model for the same compute budget by training a larger model on fewer tokens [22]. Hence, prior intuitions, such as the need for a larger batch size in the convergence phase, may not apply. For this reason, in the rest of this section, we cover works that focus on modern language model training specifically.

Zhang et al. [41] and Shallue et al. [34] examine the critical batch size, which represents the threshold beyond which greater data parallelism would lead to diminishing returns, and tune all optimizer hyperparameters. They argue that any batch size below the critical batch size should perform equally well when training for the same number of tokens. Differently from these works, we empirically demonstrate that small batch sizes, including the extreme value of batch size one, perform on par with or even better than larger batch sizes, and we also show that small batches are more robust to optimizer and hyperparameter choices and enable SGD to perform competitively. We validate these findings on language models with modern design choices and up to 1.3 billion parameters.

On the other hand, Vyas et al. [57] argue that gradient noise from small batch sizes does not yield similar performance advantages in the single epoch setting as it does in multiple-epoch training. Experimental results from Filatov et al. [58] also suggest that small batches do not work as well as large batches for language models trained on C4 [59], but notably, the authors do not tune the decay rates $\beta_1$ and $\beta_2$ of the Adam optimizer. In the same vein, Xiao [14] claim that both excessively small and large batch sizes underperform for language model training with Adam, leading to a U-shaped curve of the loss as a function of the batch size. We reproduce the exact experiments performed by Xiao [14] and show that their conclusions are an artifact of incomplete hyperparameter tuning; we

find that simply using a slower decay rate of the second moment estimate $\beta_2$ makes small batch sizes competitive.

Zhao et al. [19] train language models on C4 using different diagonal preconditioning optimizers, namely Adam, Lion, Adafactor with momentum, SGD with momentum, and signed SGD with momentum. They demonstrate that all these optimizers, aside from SGD which lags behind, achieve comparable optimal performance and robustness to hyperparameters. We challenge this conclusion in our work and argue that in the small batch size regime, SGD can perform on par with Adam.

Shallue et al. [34], Kidambi et al. [40] and Zhang et al. [41] find that momentum is not necessary for SGD when the batch size is small, which aligns with our findings. However, Kidambi et al. [40] focus on small-scale vision experiments, Shallue et al. [34] and Kidambi et al. [40] do not compare the performance of SGD vs. Adam, and Zhang et al. [41] claim that SGD performs significantly worse than Adam. We hypothesize this observation could be a result of Zhang et al. [41] using 64 as the smallest batch size – we test batch sizes all the way down to 1. Zhang et al. [35] highlight that the benefits of more sophisticated optimizers diminish as batch size decreases, but their theoretical results assume a convex quadratic objective and their empirical results only cover a single language modeling experiment that is limited to a two-layer transformer.

Porian et al. [42] and Zhang et al. [41] find that increasing the value of $\beta_2$ is important for small-batch training with Adam. However, their approach is based on discrete hyperparameter sweeps across $\beta_2$. In contrast, we introduce and validate a principled scaling rule that holds the second-moment half-life constant in terms of tokens. Busbridge et al. [60] propose a scaling rule for the decay coefficient of exponential moving averages similar to ours, but applied in the context of model weight averaging. Zhang et al. [44] show that there exists a threshold of $\beta_2^*$ such that when $\beta_2 > \beta_2^*$, Adam converges (this threshold decreases with batch size, i.e. the threshold is higher for smaller batch sizes). Chowdhery et al. [43] note that increasing $\beta_2$ during training leads to better performance for rare token embeddings. While this is an important insight, we provide a simpler batch-size-aware heuristic that applies even when using a constant $\beta_2$ throughout training.

## 4 Experimental Results

Recent works find that larger batch sizes and relatively sophisticated optimizers are necessary for stable language model training. For instance, Zhao et al. [19], Kunstner et al. [61], and Zhang et al. [62] observe that SGD has an extremely slow convergence speed for training language models in comparison to adaptive optimizers like Adam. Other works find that Adam performs poorly with small batch sizes [57, 14, 63, 64].

In this section, we revisit and challenge these beliefs by noting that such findings arise due to poor hyperparameter choices and specific experimental setups, for example, not adjusting the decay parameters $\beta_1$ and $\beta_2$ for Adam at small batch sizes, or using too large a batch size for SGD. We put these beliefs to the test by running a thorough grid search over optimizer hyperparameters and batch sizes for SGD, Adam, Adafactor, and Muon. As the cost of a grid search grows exponentially with the number of hyperparameters, we only perform exhaustive grid searches on a small transformer decoder-only model with 30 million active parameters following Liu et al. [65] and Xiao [14]. We use these results to propose heuristics for scaling hyperparameters across batch sizes, and we validate our findings at a larger scale by pretraining GPT-2 (124M) [66] and GPT-3 (1.3B) [17] and fine-tuning Gemma 3 (4B) [67]. In our experiments, we study batch sizes spanning more than four orders of magnitude: $\{1, 4, 16, 64, 256, 1024, \text{ and } 4096\}$.

### 4.1 Small Batch Sizes Render Sophisticated Optimizers Unnecessary

We train the 30M model on 600 million tokens from the FineWeb-Edu dataset [68], following Chinchilla scaling laws [22]. Our model adopts modern design choice, including rotary embeddings [69], QK-normalization [70], RMSNorm [71], GELU activations [72], and separate input and output embeddings. We use a context length of 512 and tokenize the dataset using the GPT-2 tokenizer. We provide additional experimental details in Appendix A.

To verify the effect of the batch size and optimizer choice on the performance of our model, we run a dense grid search over all the hyperparameters of SGD, Adam, Adafactor, and Muon. In particular,

we tune the learning rate and the decay factors ($\beta_1$, $\beta_2$) jointly for each batch size to obtain the hyperparameter configuration that provides the lowest validation loss.

The results in Figure 1a highlight that all optimizers perform best at the smallest batch size, and as the batch size increases, not only does the performance of each optimizer degrade but also the gap between different optimizers widens.

*Why do large batches require more sophisticated optimizers?* Every time our optimizer takes a step, it makes a prediction about the loss function away from the current parameter vector. When we train with a higher learning rate and take larger steps, for example in large batch training, our optimizer must make predictions about the loss function farther away from the current parameter vector. We hypothesize that taking larger steps leads to a harder prediction problem and that this harder prediction problem requires a more sophisticated or better-tuned optimizer.

*Why is momentum less necessary for small batch sizes?* Momentum can be seen as a means of dampening oscillations that occur on ill-conditioned loss functions [73]. When we take large steps, we may overshoot the minimum across short (i.e. high curvature) axes, leading to oscillations. By averaging gradients across iterations, these oscillations cancel out in the momentum term. However, when the optimizer takes small steps, as is the case in small batch training paired with a small learning rate, parameter updates do not overshoot the minimum across short axes, so there are no oscillations to damp. We illustrate this effect in Appendix B through a toy example.

**Summary**: The difference in performance between optimizers shrinks under small batch sizes. In fact, vanilla SGD without momentum performs competitively for small batch sizes.

## 4.2 Large Batch Training is Sensitive to Hyperparameters

To evaluate robustness to hyperparameters at each batch size, we perform an ablation around the optimal hyperparameters achieving the lowest validation loss for the Adam optimizer. We use the same 30M parameter model trained on the FineWeb-Edu dataset as in Section 4.1.

We report in Figure 3 the change in the loss with respect to the lowest value of the loss achieved at every batch size, ranging from 1 to 4096. On the x-axis, we report the scaled value of the hyperparameters relative to the value of the hyperparameter that achieved the best loss. We perform this rescaling to make the plots for different batch sizes easier to compare, since the scale of their optimal hyperparameters varies.

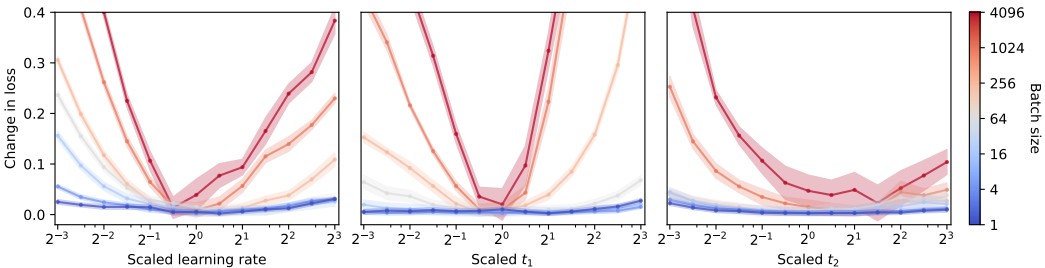

Figure 3: **Small batch sizes are robust to hyperparameter misspecification.** Loss sensitivity to learning rate, $\beta_1$, and $\beta_2$ using Adam at batch sizes from 1 to 4096. Each curve sweeps a single hyperparameter while holding the others fixed at their optimal values. We observe that smaller batch sizes exhibit broader low-loss regions across all hyperparameters, indicating greater robustness to misspecification. The x-axis shows a scaling of each hyperparameter relative to its optimum value. We parameterize $\beta_1$ and $\beta_2$ in terms of their decay half-lives, as described in Section 2. Results are shown for the 30M parameter model trained on 600M tokens of FineWeb-Edu.

We observe a striking result: small batch sizes are significantly more robust to *all* of Adam's hyperparameters than large batch sizes. More importantly, batch size one achieves a nearly optimal loss for the entire range of learning rates, $\beta_1$, and $\beta_2$ hyperparameters that we consider in our search, whereas the loss for larger batches increases sharply as we vary the hyperparameter values.

To highlight the compounding value of this robustness displayed by small batch sizes as the number of hyperparameters increases, we perform a joint 2-dimensional ablation with respect to both the learning rate and decay rate $\beta_1$ on a GPT-2 model in Figure 1b. Consistent with the previous result, batch size one yields much broader low-loss regions across both hyperparameters compared to batch size 512. Therefore, if practitioners were to consider not only the budget to run a single training run but also the computational budget for tuning the corresponding optimizer hyperparameters, a smaller batch size might be preferable.

*Why are smaller batch sizes more robust to hyperparameter choices?* Similar to the previous subsection, we hypothesize that large batch sizes require careful hyperparameter tuning since they require large step sizes and therefore have to make predictions about the loss surface further away from the current parameter vector.

**Summary**: Small batch sizes are more robust to hyperparameter misspecification and might be preferable after accounting for the hyperparameter tuning budget.

### 4.3 Adam Hyperparameters: How to Scale Them with Batch Size and Why That Is Necessary

We use our grid search results on the 30M model to test existing scaling laws for Adam hyperparameters and also devise new scaling heuristics for these hyperparameters. Figure 4 shows the validation loss on FineWeb-Edu for different Adam hyperparameters, namely the learning rate, and decay parameters $\beta_1$ and $\beta_2$. We similarly plot the loss for different values of the second moment half-life $t_2$, which refers to the number of tokens it takes for a gradient's contribution to the momentum to be reduced to half of its initial value. We provide a more detailed description of $t_2$ in Section 2.

For the learning rate, we observe that the square root scaling recommended by several works and commonly used in practice does not hold [74, 7, 37, 75, 76]. In fact, Figure 4 (left) shows that the learning rate scales more slowly than a square root factor with the batch size. For instance, as we scale the batch size from 1 to 1024, the square root rule would indicate that the corresponding learning rate should be scaled by a factor of 32, whereas we find that a factor of only about 3 empirically works better.



Figure 4: **Fixing the half-life of the second moment estimate in terms of tokens $t_2$ scales better than fixing $\beta_2$.** We plot the validation loss on FineWeb-Edu for the 30M parameter model when varying the Adam learning rate, $\beta_1$, $\beta_2$, and the second moment half-life $t_2$, across different batch sizes. The learning rate does not follow the commonly recommended square root scaling with batch size. We also observe that the default $\beta_1 = 0.9$ performs well across batch sizes. In contrast, keeping $\beta_2$ fixed leads to suboptimal performance at small batch sizes. Instead, fixing the second moment half-life $t_2$ across batch sizes provides a simple and effective scaling rule.

We also validate that the default value of the first moment decay $\beta_1 = 0.9$ used by practitioners for language model training does indeed achieve good performance across batch sizes. In contrast, keeping $\beta_2$ fixed to its default value, typically in the range $[0.95, 0.98]$ for language models, does not scale effectively to smaller batches, which require a significantly larger value of $\beta_2$. On the other hand, measuring the decay of the second moment in tokens through the half-life $t_2$ offers a strong scaling heuristic: keeping $t_2$ fixed to its optimal value as we scale up or down the batch size provides good performance without re-tuning this hyperparameter at new batch sizes. This finding translates to the following scaling heuristic: if we were to scale the batch size from $B$ to $B^*$, the new $\beta_2^*$ would depend on the old $\beta$ associated with $B$ as follows:

$$\beta_2^* = \beta_2^{(B^*/B)} \qquad (2)$$

Next, we test our scaling heuristics in a different setting proposed in Xiao [14]. The authors train a decoder-only transformer with 19M non-embedding parameters using Adam on the C4 dataset [59], and only tune the learning rate with $\beta_1$ and $\beta_2$ fixed to their default values of 0.9 and 0.95, respectively. We reproduce their Figure 10(a) results in Figure 5 (left), where small batch sizes of 16 and 32 perform worse than larger batch sizes of 128 and 256.

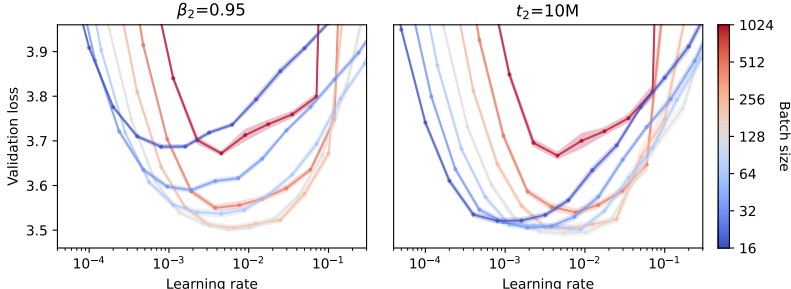

Figure 5: **Scaling $\beta_2$ according to our heuristics significantly improves the performance of small batch sizes. Left:** Results from Xiao [14], showing suboptimal performance for small batch sizes due to fixed Adam hyperparameters. **Right:** Our results after scaling $\beta_2$ such that the decay half-life $t_2$ is constant across batch sizes. The smallest batch sizes now perform on par with larger batch sizes.

While our scaling heuristics imply that $\beta_1 = 0.9$ could work well across batch sizes, we recommend scaling $\beta_2$ such that $t_2$ is fixed. By following this simple one-shot scaling recommendation and without any additional hyperparameter tuning, we achieve significantly different results in Figure 5 (right). Strikingly, small batch sizes 16 and 32 achieve on par performance with larger batch sizes 128 and 256, therefore implying that: (i) the previous results from Xiao [14] are largely a consequence of incomplete hyperparameter tuning, (ii) our simple one-shot heuristic generalizes to other settings.

> **Summary**: While keeping $\beta_1$ fixed to the default value of 0.9 scales well with the batch size, $\beta_2$ should be scaled such that the token half-life is fixed. With this prescription, small batch sizes can perform on par with or better than larger batch sizes.

### 4.4 SGD Performs Competitively with Adaptive Optimizers at the Billion+ Parameter Scale

Through our experiments with 30M parameter models, we conclude that: (1) small batch sizes can perform on par with or better than larger batch sizes, (2) small batch sizes require less sophisticated optimizers and are more robust to hyperparameter misspecification, (3) in particular, vanilla SGD can be competitive with adaptive optimizers in the small batch regime, and (4) we propose a new scaling heuristic for $\beta_2$ such that the half-life $t_2$ is fixed, and validate that it leads to improved performance. In Figure 6, we test these findings at a larger scale by training GPT-2 (124M) [66] and GPT-3 (1.3B) [17] on the FineWeb dataset [68].

For the larger 1.3B model, hyperparameter tuning of the learning rate for each optimizer was computationally prohibitive; therefore, we used AdamW with the default hyperparameter settings from Brown et al. [17] (batch size 512, $\beta_1 = 0.9, \beta_2 = 0.95, w = 0.1$) as our baseline. When using batch size 1 with Adam, we do not perform any hyperparameter tuning – we turn off weight decay and scale the learning rate following our empirical results from Section 4.3. We compare two scaling heuristics for $\beta_2$: keeping $\beta_2$ fixed and keeping the half-life $t_2$ fixed. We do, however, sweep over four learning rates for Adafactor and SGD, since we have no reference values for these optimizers. Interestingly, we find that SGD performs on par with the AdamW baseline, and Adam and Adafactor with batch size 1 both outperform the AdamW baseline.

For the smaller (124M) model, we follow the same procedure, except we tune the learning rate of each optimizer, which improves the performance of the AdamW baseline.

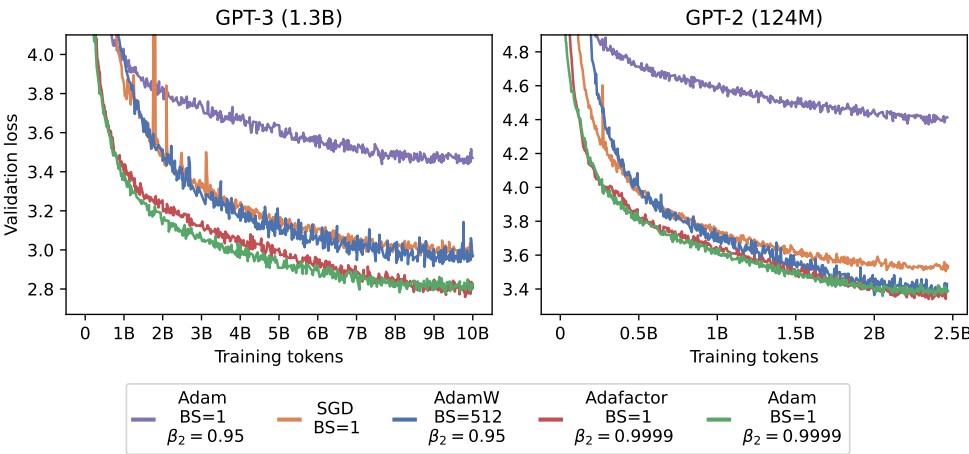

Figure 6: **Vanilla SGD performs competitively at larger model scales with minimal tuning.** We train GPT-3 (1.3B) and GPT-2 (124M) on FineWeb and compare validation loss across optimizers. **Left:** For GPT-3 (1.3B), vanilla SGD with batch size 1 and no momentum performs on par with the AdamW configuration from Brown et al. [17], despite having no optimizer state. Our proposed $\beta_2$ scaling rule improves Adam's performance at small batch sizes and outperforms the baseline. **Right:** After tuning the learning rate of each optimizer on GPT-2 (124M), Adam and Adafactor at batch size 1 match the performance of AdamW at batch size 512, while SGD performs slightly worse.

## 4.5 Full Fine-tuning on a Budget

In Sections 4.1 to 4.4, we studied pretraining across different model scales and discovered that with small batch sizes, optimizers with small or even no state (like Adafactor and SGD) can perform competitively with Adam. We now exploit this observation for memory-efficient fine-tuning.

While pretraining is typically bottlenecked by compute, fine-tuning is instead often bottlenecked by memory. Indeed, fine-tuning a large model on a small dataset does not require many FLOPs, but it does require storing the (large) model and optimizer state in memory. In Figure 7, we show an example where the GPU has enough memory to store the model, but not enough memory to also store an expensive optimizer like Adam.

A common approach to reduce memory usage during fine-tuning is to freeze the model weights and only train Low-Rank Adaptation (LoRA) [77] modules on top of the frozen model, which substantially reduces the number of trainable parameters and also proportionally reduces the size of the optimizer state. However, LoRA often underperforms full parameter fine-tuning [78, 79].

We propose to use a small batch size combined with a small optimizer like Adafactor to get the performance benefits of full fine-tuning while having similar memory requirements to LoRA.

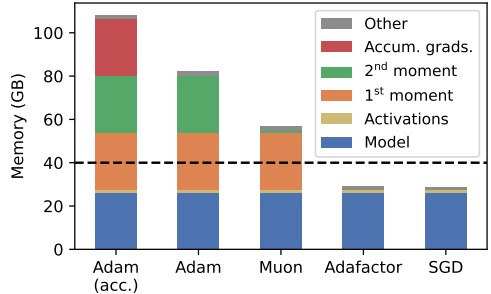

Figure 7: **Minimum memory requirements** to train GPT-3 (13B), assuming all parameters and activations are stored in 16-bit precision, gradient checkpointing is used after every transformer block layer, and the backward pass is fused with the optimizer update step (so that the full model gradient never needs to be materialized in memory). The dashed line shows the available memory on an NVIDIA A100 40GB GPU, which is not large enough to store expensive optimizer states or accumulated gradients. We measured the true memory usage of each optimizer by performing fused backward passes in PyTorch on an NVIDIA B200 GPU.

We provide an example in Figure 8 by fine-tuning a non-instruction-tuned Gemma 3 (4B) on the MATH dataset for 5 epochs. As a baseline, we fine-tune the model with Adam and batch size 16 (requiring gradient accumulation), storing all parameters in float32 precision, which requires storing a total of 16 bytes in memory per model parameter. Another baseline is provided by LoRA, which requires only around 2 bytes per model parameter since the pretrained model checkpoint is stored in bfloat16 precision, and the number of trainable parameters for LoRA is negligible compared to the model size.

We make two observations based on our results. First, when using Adam, we can reduce the batch size all the way down to one while following the $\beta_2$ scaling heuristic from Equation (2), eliminating the need to store accumulated gradients in memory. Second, when a small batch size is used, we can replace Adam with Adafactor, which has a much smaller state size. To further reduce the memory use of Adafactor, we store its (tiny) optimizer state in float32, but we store model weights in bfloat16, using stochastic rounding to update the weights after every optimizer step (we discuss the importance of stochastic rounding in Appendix A.3).

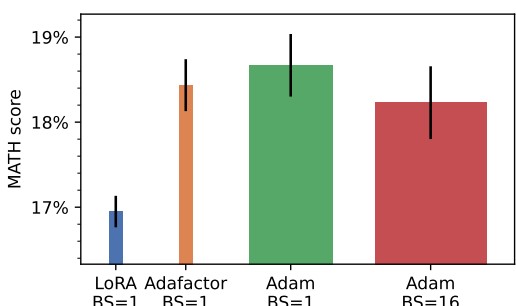

Figure 8: **Adafactor enables memory-efficient fine-tuning**. We fine-tune and evaluate a non-instruction-tuned Gemma 3 (4B) model [67] on the MATH dataset [80]. We compare the use of LoRA ($r = 16$) against full model fine-tuning, and a small vs. large batch size. For each training method, the width of the plotted bar is proportional to the memory footprint. LoRA uses bfloat16 model weights and float32 adaptors, and Adafactor similarly uses bfloat16 weights but float32 optimizer state. We fix the second moment half-life to be consistent across batch sizes ($\beta_2 = 0.95$ for batch size 16 and $\beta_2 = 0.997$ for batch size 1). We run each optimizer for 5 epochs and sweep over 8 learning rates. We train LoRA using Adam.

**Summary**: We recommend that practitioners training large models in memory-constrained settings exploit the benefits of small batch sizes (e.g. using optimizers with a small state size) rather than trying to emulate the large batch size setting typically used in industry.

## 5   Practical Recommendations and Discussion

We show that small batch sizes, even as low as 1, consistently match or outperform large batches across model scales when tuned properly, in both pretraining and fine-tuning. In particular, SGD with batch size 1 performs on par with the AdamW baseline for GPT-3 (1.3B) with minimal tuning. We also propose a simple scaling rule for $\beta_2$ based on a fixed second-moment half-life in tokens, which generalizes across model scales. Our results contradict common assumptions about large batch training and gradient accumulation.

Our general recommendation for setting the batch size is to **use the smallest batch size that maximizes model throughput** (measured in tokens / second) or equivalently the model FLOPs utilization (MFU) [43]. To achieve this objective, the accelerator must spend more time performing computation than moving data between HBM and register memory, which in practice means using a batch size of at least several hundred tokens per device [81]. When training frontier models on large clusters, this can correspond to a batch size of many millions of tokens. But when training on a single device, the smallest batch size required to achieve high arithmetic intensity could be as low as a few thousand tokens. We therefore recommend **avoiding gradient accumulation** unless training on multiple devices with multiple model replicas, where the bandwidth between model replicas is a bottleneck. Lastly, when working in a memory-constrained setting, we recommend **exploiting the benefits of simple optimizers combined with a small batch size**.

Several open questions remain. How do these findings interact with batch size schedules? Can we further reduce weight precision in fine-tuning from 16 to 8 or even 4 bits? Can we design more effective optimizers with a small state, tuned to the small batch size regime? And can we theoretically understand why fixing the second-moment half-life in tokens generalizes across scales?

## Acknowledgements

We thank Shikai Qiu and Alexandra Souly for helpful discussions. This work was supported by NSF CAREER IIS-2145492, NSF CDS&E-MSS 2134216, NSF HDR-2118310, BigHat Biosciences, and Google's TPU Research Cloud (TRC) program: https://sites.research.google/trc/.

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

# A    Experimental Details

## A.1    Model architecture

Across all of our pretraining experiments, we used four decoder-only transformer models [82] that share the same architecture and only differ in layer dimensions and tokenizer. We specify the exact layer dimensions and tokenizers used in Table 1.

We used a model architecture based on GPT-2 [66] with several small changes based on modern practices to improve model FLOPs utilization (MFU) [43], simplify implementation, and marginally improve model performance:

- Head dimension 128
  - We increased the head dimension on the smaller models from 64 to 128 to achieve a higher MFU. We trained each model on TPU v4, which has 128 x 128 multiply-accumulators. Using a head dimension any lower than 128 would prevent us from filling the multiply-accumulators, thereby lowering MFU. This decision follows the Gemma family of models optimized for TPU training [67].
- RMSNorm [71]
  - We wanted to include the Muon optimizer in our benchmarks, but Muon does not support training 1D layers. To get around this limitation, we simply omitted the use of any 1D layers by using RMSNorm with no bias.
- Untied embeddings
  - We do not tie the weights of the first and last layer. This increases the number of trainable parameters but does not increase the number of active parameters or the cost of training.
- Rotary positional embeddings (RoPE) [69]
- QK-norm [70]
- GELU [72]

We pretrained most models to follow Chinchilla scaling laws, i.e. using 20 tokens per active parameter. The only exception to this rule is the GPT-3 (1.3B) model, which we only trained for 10B tokens, similar to the original GPT-2 (1.5B) model.

| Model | 30M | 19M[14] | GPT-2 (124M) | GPT-3 (1.3B) |
|---|---|---|---|---|
| Dataset | Fineweb-Edu | C4 | Fineweb | Fineweb |
| Tokenizer | GPT-2 | T5 | GPT-2 | GPT-2 |
| Vocabulary size | 50257 | 32101 | 50257 | 50257 |
| Model / embedding dimension | 384 | 512 | 768 | 2048 |
| Hidden dimension | 1536 | 2048 | 3072 | 8192 |
| Head dimension | 128 | 128 | 128 | 128 |
| Number of layers | 6 | 6 | 12 | 24 |
| Sequence length | 512 | 512 | 1024 | 2048 |
| Non-embedding parameters | 11M | 19M | 85M | 1.2B |
| Embedding parameters | 2 x 19M | 2 x 16M | 2 x 39M | 2 x 103M |
| Active parameters | 30M | 35M | 124M | 1.3B |
| Total trainable parameters | 49M | 52M | 162M | 1.4B |
| Training tokens | 600M | 705M | 2.5B | 10B |

Table 1: Model dimensions and configurations across different architectures.

## A.2    Learning rate schedule

For pretraining, we used a linear warmup for the first 5% of the training steps (from zero to peak learning rate), followed by cosine decay (from peak learning rate to zero). We use the same schedule for all batch sizes. For fine-tuning, we used a constant learning rate.

## A.3 Stochastic rounding

Across all experiments, we compute activations and store the optimizer state in float32 precision. We also store all trainable model parameters in float32 precision *except* for Adafactor in Figure 8, which uses bfloat16 weights in order to match the memory footprint of LoRA. While it is a common practice to use bfloat16 weights for computation, the master copy of model weights is typically still stored in float32 to enable accumulating small updates to the weights. When the master weights are stored in bfloat16 (which only has ∼2.4 decimal points of precision), standard deterministic rounding can bias the weight updates, meaning that accumulating gradients across many small batches no longer approximates computing gradients from a single larger batch. Because of this bias, reducing the batch size might actually be detrimental with standard deterministic rounding. One approach to remove this bias is to compute updated weights in float32, then stochastically round them to bfloat16 for storage. In a pretraining experiment in Figure 9, we show that stochastic rounding

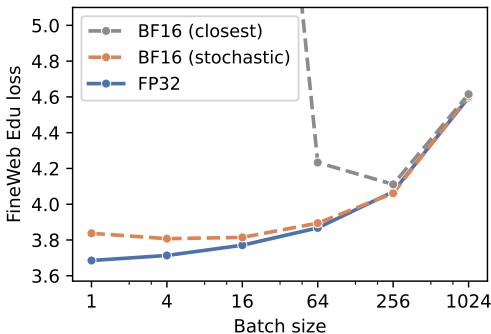

Figure 9: **Stochastic rounding enables bfloat16 training.** We repeat the training runs from Figure 1a using the optimum hyperparameters found for Adafactor. We ablate the precision of the model weights, using: (1) float32 – same as in Figure 1a, (2) bfloat16 with closest rounding, (3) bfloat16 with stochastic rounding. The optimizer state is always stored in float32 precision since it has a negligible size.

can bring the performance of bfloat16 weights close to the performance of float32 weights, even for the smallest batch sizes. In the fine-tuning experiment in Figure 8, we did not observe a statistically significant difference between float32 weights and bfloat16 weights with stochastic rounding for Adafactor.

## A.4 Computational cost

We used 32 TPU v4 chips to train all models [83]. Table 2 shows the time for a single training run of each model as well as the total computational cost for each model. Using batch sizes 1 and 2 resulted in roughly a 30-70% drop in MFU compared to using a larger batch size, depending on the model size and sequence length. In practice, we do not recommend using a batch size so small that it severely degrades MFU. We only used the smallest batch sizes to scientifically study the effect of batch size.

| Model | 30M | 19M[14] | GPT-2 (124M) | GPT-3 (1.3B) | Gemma3 (4B) |
|---|---|---|---|---|---|
| Accelerator | 1 TPU v4 chip | 1 TPU v4 chip | 1 TPU v4 chip | 4 TPU v4 chips | 4 TPU v4 chips |
| Average run time | 24 min | 25 min | 7.5 hours | 100 hours | 44 min |
| Num. runs: Figure 1 | 3110 | / | 312 | 11 | / |
| Num. runs: Figure 3 | 1911 | / | / | / | / |
| Num. runs: Figure 4 | 6500 | / | / | / | / |
| Num. runs: Figure 5 | / | 1260 | / | / | / |
| Num. runs: Figure 6 | / | / | 40 | / | / |
| Num. runs: Figure 8 | / | / | / | / | 256 |
| Total num. of runs | 11521 | 1260 | 352 | 11 | 256 |
| Total TPU v4 chip hours | 4600 | 525 | 2640 | 4400 | 750 |

Table 2: Computational cost of each experiment.

## B Additional Results

In Figure 10 we illustrate on a toy problem why momentum is required when the batch size is large but not required when the batch size is small. We run SGD on two variables $(x, y)$ to minimize the value of a loss function defined as $x + 10y^2$. Notice that this loss function is much steeper in the vertical direction compared to the horizontal direction.

We simulate the use of a small and a large batch size by adding noise to the gradient estimate at every step. In particular, we define the minibatch gradient estimate at each step to be the true gradient

multiplied by a random variable sampled from a normal distribution $\mathcal{N}(1, \sigma^2)$, where the noise scale $\sigma$ controls the signal-to-noise ratio of the minibatch gradient estimate. We simulate a large batch size by using a high signal-to-ratio of 5, and a small batch size by using a small signal-to-noise ratio of only 0.3.

In our experiments training language models, we fixed the computational budget across all batch sizes by only training for a single epoch. Analogously in this toy example, we only run 10 optimization steps using the simulated large batch size, but we run 100 optimization steps using the small batch size. The idea is that if we use a small batch size, the minibatch gradient estimates at each step become more noisy, but in turn we get to take more optimizer steps.

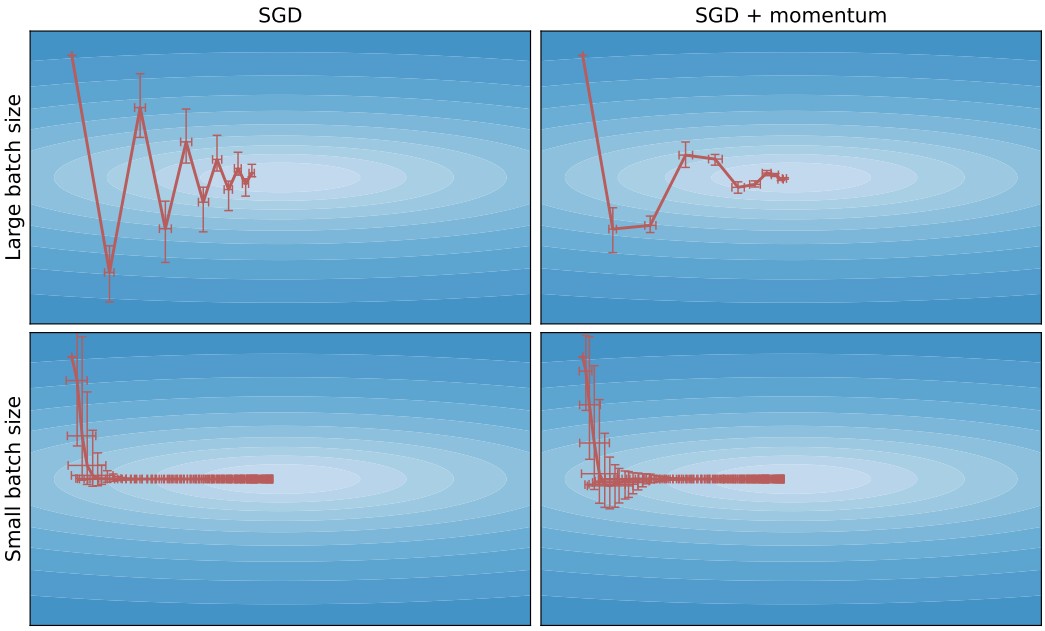

Figure 10: **Toy optimization problem.** We compare the use of SGD and SGD with momentum on a loss function defined as $x + 10y^2$. We simulate a small and a large batch size by adding Gaussian noise to the gradient estimate at each optimizer step. The error bars show a 50% interquartile range sampled across different random seeds for the minibatch gradient noise.

In the top row of Figure 10, we see that when a large batch size is used, we only get to take a small number of steps, and so in turn every step has to be large. Because the loss function is much steeper in the vertical direction compared to the horizontal direction, the large step size results in oscillations along the vertical direction. In this case, using momentum helps dampen the oscillations along the vertical direction and speeds up convergence along the horizontal direction. In fact, in the extreme case of full-batch gradient descent (which has no gradient noise), using momentum probably improves the convergence rate of gradient descent for this toy problem [73].

We show the small batch size setting in the bottom row of Figure 10. Notice that since the batch size is small, every step becomes more noisy, but in turn we can take more steps and reduce the size of each step. As a result of decreasing the step size, the optimizer no longer overshoots along the vertical direction, obviating the need for momentum. There are no longer any oscillations to dampen, so using SGD with and without momentum results in similar performance.

