# OpenReview forum: "Small Batch Size Training for Language Models: When Vanilla SGD Works, and Why Gradient Accumulation is Wasteful"
_NeurIPS.cc/2025/Conference — NeurIPS 2025 poster_

### Official Review · Reviewer_DoJ4 · 2025-06-07

**Clarity:** 4
**Significance:** 2
**Originality:** 2
**Rating:** 5
**Confidence:** 5

**Summary:**

The paper is interested in training with small batch sizes.  By ML theory (McCandlish et al), you'd expect that smaller batches are asymptotically more and more compute efficient (able to reach the same loss with less data), but recent practice (e.g., Porian et al.) has found there is an "optimal batch size" below which performance degrades.  Others have speculated that the problem with small batches might be that we should re-tune Adam's Beta parameters (Porian et al., Zhang et al.).  In this paper, they show that if, instead of holding β2 fixed, you hold its "half-life" fixed, in tokens (key point), then you can train stably all the way down to B=1.  This scaling rule for β2 essentially means that if you increase the batch size by a factor of K, we should scale optimal β2 by β2^K.  Interestingly, even at B=1, vanilla SGD without momentum performs somewhat reasonably.  This echos other work suggesting that a lot of the advances in optimizers (e.g., Muon, AdamW weight decay, LR decay schedules) are just methods to deal with gradient variance, but another way to deal with it is to just take a lot of very small steps.  Unfortunately, the paper doesn't test using AdamW, only vanilla Adam, and they don't reveal the LR schedule, so we don't really know how their recommendation plays with SOTA LLM training setups.

**Questions:**

- What is the LR schedule?

- Intuitively, why would it be the case that we should maintain the velocity \beta_2 timescale but not the momentum \beta_1?

- The paper recommends making B so small that it maximizes model throughput.  Do you mean the global optimum?  Because then we could just as easily say, "set the B to the LARGEST value that maximizes model throughput" or "the single value that maximizes model throughput."  Perhaps you are really looking at some kind of tradeoff between loss and throughput?  That is, if B is too large, then loss will be worse, and it may take longer to train.  So we could frame this as finding the point where we can get the lowest loss in the least amount of wall-clock time, given a fixed amount of compute resources?  This is actually a good contrast case from the work in Bergsma et al (2505.13738), which assumes you want to minimize wall-clock time but have infinite parallel resources.  Furthermore, some kind of experimental results in this section would have been nice.  E.g., you could actually show that following your recommendations does improve a real training run.

**Ethical Concerns:**

["NO or VERY MINOR ethics concerns only"]

**Final Justification:**

This is interesting work and the authors did a very reasonable job.  After reading the rebuttal, I felt like my main addressable concerns were addressed.  I read the other reviews and feel like Reviewer uKGs and myself are most sympatico, and believe they may also raise their score after the rebuttal.

The "Accept" rating means: "Technically solid paper, with high impact on at least one sub-area of AI or moderate-to-high impact on more than one area of AI, with good-to-excellent evaluation, resources, reproducibility, and no unaddressed ethical considerations."
- I am not 100% sure about "high impact" on the field of optimization, but overall this description of the paper most resonates with me.

**Limitations:**

Really only a very cursory discussion of limitations in the conclusion, and no real discussion of societal impact (good or bad!).  Having a limitations section, even in the appendices, would have been very nice, because it would have also helped the reviewer/reader understand what exactly was done here: did you use weight decay?  Did you use a LR schedule?  Did you try overtrained models in addition to Chinchilla-optimal?  Did you try very large models?  Etc.

**Quality:**

3

**Strengths And Weaknesses:**

# Strengths

## Quality and clarity:

- Good writing, great organization, good experiments, great figures: easy to follow, easy to evaluate, and easy to read.
- The idea of keeping the timescale of \beta_2 fixed is intuitive and interesting and cool.
- The claims are well-supported.

## Significance and originality:

- The scaling law for \beta_2 in particular is simple to implement, so could be beneficial to small-batch training runs, although the motivation for such small-batch (pre-training) runs is less clear.  I have not seen this particular scaling law for \beta_2.

# Weaknesses:

## Discussion of prior work

- Many of the key findings have been observed previously in other contexts, in papers that are not cited or cited for different reasons.
- Plus, there is work that is very related.
- I worry one of the other reviewers will be an author of one of these papers and they're going to UNLOAD on you :)

Here are some pointers:

Shallue et al. 1811.03600v3
- SGD without momentum can be competitive at small batch sizes: "However, at batch sizes small enough that all optimizers are within their perfect scaling region, momentum optimizers perform identically to SGD without momentum. Though initially surprising, this identical performance at small batch sizes is consistent with observations made in Kidambi et al. (2018)."

Zhang - How does critical batch size scale in pre-training - 2410.21676v1:
- SGD without momentum can be competitive at small batch sizes: "We observe that in small batch size regimes like 2^6, the performance gap between optimizing with and without momentum β1 is small while the gap increases as we double the batch size (Shallue et al., 2019)."
- Tuning \beta_2 is crucial for AdamW at small batch sizes: "found that β2 in Adam, the exponential decay rate of the second momentum estimate of gradients to smooth out the model update, also has significant effects on training for small batch sizes. This might be because gradients in small-batch training are sparser. Specifically, we ablate β2 ∈ [0.95, 0.99, 0.999] for all model sizes and batch sizes in [64, 128, 256, 512]. We find that the default value 0.95 in previous works that are set for millions of tokens batch size training might be sub-optimal"

Porian et al., 2406.19146v3
- Tuning \beta_2 is crucial for AdamW at small batch sizes: "we observe that setting the AdamW β2 parameter to be 0.95 is suboptimal at smaller batch sizes (128 and below), and increasing it allows establishing clear trends from our small-scale hyperparameter sweep"
- "Initially, we kept β2 at its previous value of 0.95. However, this led to poor results at smaller batch sizes: as the batch size gets smaller, the squared gradients become noisier, and AdamW requires more smoothing to obtain a correct denominator. Therefore, we added 0.99 and 0.999 to the sweep, obtaining improved performance on small batch sizes. This empirical observation matches the theoretical work by Zhang et al. [63], that showed that when the batch size is small, higher values of β2 are crucial for the convergence of Adam."  FYI, [63] is "Y. Zhang, C. Chen, N. Shi, R. Sun, and Z.-Q. Luo. Adam can converge without any modification on update rules. Advances in neural information processing systems, 35:28386–28399, 2022."

Busbridge - How to scale your EMA - 2307.13813v3:
- The scaling rule for β2 here is exactly the same as the scaling rule for the momentum parameter --- in a model _weight_ EMA --- in Busbridge.

- Also worth considering the EMA view of AdamW proposed by Wang & Aitchison (2405.13698) – looks at the timescale of AdamW, as a function of η and λ rather than the βs.

- Also, the AdEMAMix optimizer 2409.03137 seems related as both methods essentially try to capture "older" information.  Maybe that paper should have experimented more with different \beta_2 values?

- Related to these prior paper studying AdamW rather than Adam: see scaling weight decay with batch size, in Bergsma et al (2505.13738) as motivated by the EMA view of Wang & Aitchison (2405.13698)

Chowdhery et al - PaLM - Scaling Language Modeling with Pathways - 2204.02311
- "The second-order moment interpolation value is computed as 1-k^(-0.8), where k is the step number. We have found this to be more stable than the standard B2 when training large language models, because rare embedding tokens can have poorly estimated second moments over
shorter windows"
- Could something to do with embeddings help explain your results?

## Missing theory

Finally, this paper seems to miss that ML theory, e.g., McCandlish et al., specifically predict that smaller batches are (asymptotically) better, i.e., can reach a given target loss in fewer steps.
- this paper supports this hypothesis over the U-shaped optimal batch size, so it would be good to approach this directly in the discussion.

## Generality of findings

Mainly, my other concerns revolve around whether this is a competitive LLM training setup:
- we use Adam here rather than AdamW, but weight decay is used in all SOTA LLM training, AFAIK
- nothing is said about the LR schedule.  Decaying the LR can also reduce variance (Bergsma et al., 2502.15938), so what about that?
- models are undertrained (too large for their token budget).  I was especially confused by the statement: "We train our 49M model on 600 million tokens from the FineWeb-Edu dataset [48], following the Chinchilla scaling laws [25]."  The appendix likewise says, "We trained most models to follow Chinchilla scaling laws, i.e. using 20 tokens per active parameter." But 49M on 600M is more like 12 TPP, not 20 TPP.  The GPT-3 model of size 1.3B trained on 9B tokens, that's 6.9TPP -- that is highly under-trained.  No one would train such a model in practice because we could get a better loss with a smaller model (easier to serve) using fewer compute FLOPs.
- you do go as high as 20TPP in one set of experiments, but overtrained models are also very popular these days so it would be nice to understand how your results generalize to high TPP, where gradient variance is even higher.
- also, I'm struck by the lack of motivation for small-batch training.  Small batches are unlikely to be used in training giant models.  Yes, we could use smaller batches at smaller scales, but then would we worry that those findings don't transfer as well to larger scales?  Yes, we could use small batches in FT, but this paper mostly looked at PT, right?

# Minor points:

- "As a consequence, optimizers that perform well in large-batch training are increasingly standard practice" -- Was this really the motivation behind (cited) Muon, Sophia, Shampoo, SOAP, LAMB?  I think LAMB was motivated to let us train FASTER (greater parallelism), not to address stability issues.  I never heard e.g., SOAP or Muon motivated as a consequence of the belief that large batches are more stable?

- "4.3 Adam Hyperparameters: How to Scale Them with Batch Size and Why That is Necessary" - note this section only very loosely addresses the question of *why* tuning β2 is necessary.  The only explanation in this section seems to be: because it works better.

- "we hypothesize that large batch sizes require careful hyperparameter tuning since they require large step sizes and therefore have to make predictions about the loss surface further away from the current parameter vector."  Technically speaking, a "hypothesis" is an idea that you can test.  I am not sure that's the right word to the conceptual intuition given here.

- Line 105, just for completeness, should specify "where d1 and d2 are ..."

---

> ### Author Rebuttal · Authors · 2025-07-31
>
> Thank you for your thoutghful feedback. We address the points you raised below.
>
> **Extended related work discussion:** Thank you for providing pointers to related work. We have now updated our manuscript to include the following discussion of how our work extends and contrasts with these previous works.
>
> [1] and [2] report that momentum provides limited benefits when batch sizes are small. These results are aligned with our findings and validate our empirical observations. However, [1] does not compare the performance of SGD vs Adam and [2] claims that SGD performs significantly worse than Adam. We hypothesize this observation could be a result of [2] using 64 as the smallest batch size. We test batch sizes all the way down to 1.
>
> [3] similarly show that increasing the value of $\beta_2$ is important for small-batch training with AdamW. However, their approach is based on discrete hyperparameter sweeps across a few values of $\beta_2$. In contrast, we introduce and validate a principled scaling rule that holds the second-moment half-life constant in terms of tokens. This leads to a predictable and interpretable formula for scaling $\beta_2$ with batch size. We validate this rule across different optimizers, model sizes, and datasets, and show that it leads to better performance in a zero-shot manner without sweeping.
>
> [4] propose a similar token-timescale interpretation of exponential moving averages in the context of model weight averaging. While their work supports the intuition behind our $\beta_2$-scaling rule, our focus is on second-moment estimation and we validate the scaling law across a broad range of pretraining and fine-tuning settings for LLMs.
>
> The AdEMAMix optimizer [5] also addresses long-range information retention in gradient estimation by mixing momentums with different timescales. However, our contribution shows that such complexity may not be necessary: by simply tuning $\beta_2$ using our scaling rule, even Adam can perform competitively at small batch sizes. Unlike AdEMAMix, our recommendations apply to widely used, off-the-shelf optimizers and are validated empirically in memory-constrained fine-tuning settings.
>
> [6] and [7] provide a more theoretical perspective on how optimizer behavior and weight decay can be interpreted in terms of timescale or EMA dynamics. Our contribution complements theirs by operationalizing this perspective into a concrete $\beta_2$ scaling rule and empirically verifying it across different LLM training configurations. We also show that this scaling rule is critical for enabling training with batch size 1, something not addressed in their work.
>
> [8] note that dynamically adjusting $\beta_2$ leads to better performance for rare token embeddings. While this is an important insight, our scaling rule provides a simpler, batch-size-aware heuristic that applies even when using constant $\beta_2$ throughout training.
>
> References:
> - [1] Shallue et al. - 1811.03600v3
> - [2] Zhang et al. - 2410.21676v4
> - [3] Porian et al. - 2406.19146v3
> - [4] Busbridge et al. - 2307.13813v3
> - [5] AdEMAMix optimizer - 2409.03137v2
> - [6] Bergsma et al. - 2505.13738v1
> - [7] Wang & Aitchison - 2405.13698v3
> - [8] PaLM - 2204.02311v5
>
> **Generality of our findings and new results:** In the submitted version of our paper, we pretrained models ranging from 30M active parameters to 1.3B. We pretrained all models (except the largest 1.3B model) with 20 TPP. We trained the 1.3B model for 10B tokens following the GPT-2 paper as well as the modded-nanoGPT repository, purely because we were compute-constrained. Since then, we have expanded our results in two ways. First, we tested more optimizers at the 124M and 1.3B scales.
>
> |               |  AdamW, BS=512 (baseline) |  Adam, BS=1, ${\beta_2}=0.95$ |  Adam, BS=1, ${\beta_2}=0.9999$ |    Adafactor, BS=1 |  SGD, BS=1 |
> | ---                    | ---                  | ---                       |  ---                | ---                    | ---                  |
> | GPT-2 (124M)    | 3.43           | 4.40                |  3.38               | 3.36⭐️                   | 3.52            |
> | GPT-3 (1.3B)      | 2.95               | 3.45              |  2.79            | 2.78⭐️              | 2.97                  |
>
> Second, we added fine-tuning. We fine-tuned a non-instruction-tuned Gemma3 4B on the MATH dataset for 5 epochs and measured its accuracy on the validation question set. We found Adafactor (in bfloat16 precision with stochastic rounding) and batch size 1 to perform just as well as Adam in float32 precision with batch size 1 or 16. We believe this experiment is important both to prove the generality of our findings but also to provide a memory-efficient full fine-tuning recipe as an alternative to LoRA.
>
> | Adam, BS=16 | Adam, BS=1 | Adafactor, BS=1 | LoRA, BS=1 |
> | ---                    | ---                  | ---                       |  ---                |
> | 18.2 ± 0.4%    | 18.7 ± 0.4%   | 18.7 ± 0.2%        | 16.7 ± 0.4% |
>
> **Practical motivation:** We believe that this work provides at least two practical contributions. **First,** we show that smaller batch sizes are much more **robust to hyperparameter** values and optimizer design, meaning that without any hyper-parameter tuning or with limited hyper-parameter tuning, smaller batch sizes should work better in expectation compared to larger batch sizes. It is a common practice to use gradient accumulation to simulate larger batch sizes. We posit that accumulating gradients not only increases memory consumption, it also leads to worse performance. **Second**, we extend our findings to fine-tuning in a memory-constrained setting. Because it is commonplace to fine-tune large models on small datasets, fine-tuning is very often memory-constrained rather than compute-constrained. We show that in a memory-constrained setting, it is possible to use a small batch size combined with a simple optimizer like Adafactor to get the performance benefits of full fine-tuning while having little memory overhead compared to just storing the model in GPU memory, i.e. requiring only around 2 bytes / parameter.
>
> **Chinchilla scaling:** We used Chinchilla scaling as in "20 tokens per (active) parameter". We only used dense models for all experiments but we did not tie the input and output embeddings, which resulted in a different number of total vs. active parameters. For example, the "49M" model only has 30M active parameters, which is why we trained it on 30M x 20 = 600M tokens. We specify the exact dimensions of each model in Table 1 in the Appendix.
>
> **Learning rate schedule:** We used a linear warmup for the first 5% of the training steps (from zero to peak learning rate), followed by cosine decay (from peak learning rate to zero).
>
> **AdamW:** AdamW is indeed more widely used in practice than plain Adam. Unfortunately, weight decay introduces an additional hyperparameter whose optimal value depends on the batch size. When comparing different optimizers in Figure 1a, we wanted to perform a dense grid search over all hyperparameters for all batch sizes to ensure the comparison is as rigorous as possible. We did not want to use any scaling heuristics for weight decay, as this could bias the results. Since we could not afford to increase the dimensionality of the dense grid search from 4 to 5 dimensions, we did not use weight decay in this experiment at all. Naturally this opens up the question: would the results look any different for AdamW, compared to Adam? While this is not something we measured directly using a grid search, it is something we tested indirectly. Namely in Figure 5, the baselines for GPT-2 and GPT-3 actually do use weight decay, so their labels should be "AdamW", not "Adam". Hence this result provides a partial answer: Adam with batch size 1 works at least as well as AdamW with batch size 512. AdamW with batch size 1 could only possibly work better, if we tuned the weight decay coefficient instead of fixing it to zero.
>
> **Why fixing $\beta_2$ half-life works:** We illustrate in the Appendix using a toy example why the first moment is not necessary for small batch sizes, but it also does not hurt performance. As the batch size is increased, momentum starts to become necessary for good performance, however, $\beta_1$ cannot be set too high, because then the optimizer step direction becomes insensitive to local gradient information. In contrast, we hypothesize that the role of the second moment is to provide preconditioning at a longer token time-scale. We do not have a theoretical foundation to motivate these claims but we believe that our empirical results are well-supported and could be of practical value to the community. We like your intuition that the second moment needs a longer token time-scale to capture rare tokens.
>
> **Throughput:** Our general recommendation is to use the smallest batch size that results in good hardware utilization. Modern ML accelerators typically require an arithmetic intensity of at least several hundred FLOPs / byte (i.e. several hundred tokens in a batch per active model parameter) to saturate their matrix multiplication units. We expect that increasing the batch size beyond this point provides little additional benefits in terms of hardware utilization, even if the accelerator has enough memory to work with a larger batch size. Hence we recommend using the smallest batch size that achieves optimal or near-optimal hardware utilization. We share a table in our response to rewviewer 3pDd that shows the model throughput we were able to achieve for different batch sizes, although this is very much hardware / implementation specific.
>
> Thank you again for your thoughtful feedback. We believe the extended related work discussion, clarifications and additional experiments meaningfully improve our paper, and we hope you will consider raising your score. Please let us know if you have additional questions we can address.

---

> ### Comment · Reviewer_DoJ4 · 2025-07-31
> **Response to rebuttal**
>
> I have read the rebuttal to my original review and I find the following points compelling:
> - willingness to expand their discussion of related work, demonstrating good grasp of it ✔️
> - interesting new experiments with other optimizers and FT ✔️
> - convincing discussion of practical motivation ✔️
> - clarifications on other points that I raised (which I gather will be used to "meaningfully improve [the] paper") ✔️
>
> I will revise my score.

---

### Official Review · Reviewer_uKGs · 2025-06-30

**Clarity:** 3
**Significance:** 3
**Originality:** 2
**Rating:** 5
**Confidence:** 4

**Summary:**

The paper argues that neural networks can be trained in a stable fashion at small batch sizes, with non-preconditioned optimizers like SGD, even without the use of momentum. It does experiments on various scales showing that the gap between preconditioned and non-preconditioned optimizers decreases at small batch sizes. It also shows on small scale experiments that training at small batch sizes is much more robust to hyperparameters as compared to large batch sizes.

**Questions:**

1. One major thing missing from the current manuscript is the description of the learning rate schedule. Can the authors please add a proper description and how it is varied with batch size.


2. In Section 5.2, about hardware utilization, the authors state that we need a minimum batch size of a few hundred tokens to not be bottlenecked by memory bandwidth. Are the authors referring to the memory bandwidth of the GPU? Is this about transferring the data to the GPU or between HBM and SRAM of the GPU? Can the authors make it more clear?

**Ethical Concerns:**

["NO or VERY MINOR ethics concerns only"]

**Final Justification:**

I have looked at the rebuttals and other reviews, and with the discussion of previous related work, I consider this paper as technically solid. I am therefore revising my rating to 5.

**Limitations:**

The main limitation of the work is missing a proper discussion of previous theoretical and empirical works with similar claims. If the authors can improve on this, I would be happy to revise my rating.

**Quality:**

3

**Strengths And Weaknesses:**

The main strength of the work is the rigorous exploration of the question of training at small batch sizes. By proper sweeps, it shows that training at small batch sizes is even more robust to hyperparameters, and generally performs at par with large batch sizes. It also provides a general scaling rule for $\beta_2$ for Adam, based on keeping its half-life constant in terms of number of tokens.

In my opinion, the main weakness of the work is in not citing some previous theoretical works [1,2] which have already shown that preconditioners benefits decrease at small batch sizes. Other theoretical works [3] also establish that the benefit of momentum decreases at small batch sizes. Apart from this, scaling beta2 with small batch sizes has also been shown by previous works [4]. I do think the work is important as it highlights a crucial point - we can always train comparably at small batch sizes, even with more robustness to hyperparameters. However, I certainly think that the paper needs to properly discuss these related works and state that they empirically verify in language modeling setup that these things hold. Their practical recommendations are very useful. I would be willing to revise my rating once proper discussion of previous related work is done.

[1] - Which Algorithmic Choices Matter at Which Batch Sizes? Insights From a Noisy Quadratic Model - 2019

[2] - New Insights and Perspectives on the Natural Gradient Method - 2020

[3] - On the insufficiency of existing momentum schemes for Stochastic Optimization - 2018

[4] - Resolving Discrepancies in Compute-Optimal Scaling of Language Models 2024

---

> ### Author Rebuttal · Authors · 2025-07-31
>
> Thank you for your detailed and constructive feedback. We address your feedback below.
>
> **Related work discussion:** Thank you for highlighting important related work that we missed in our submitted manuscript. We have now properly cited these works and updated our related work section to include the following discussion of how our work complements and differs from these works.
>
> While similarly to our work, Zhang et al. (2019)[1] highlights that the benefits of more sophisticated optimizers diminish as batch size decreases, their theoretical results assume a convex quadratic objective that is quite different from language model training. Their empirical experiments on the other hand only cover a single language modeling experiment that is limited to a two-layer transformer. We believe that the training dynamics and optimization differ significantly for large language models and our purpose is to provide practical recommendations for LLM training in particular. Martens (2020)[2] establish theoretical benefits of second-order preconditioners. However, they do not study the effect of batch size on the performance of different optimizers, nor do they focus on LLM optimization.
>
> Kidambi et al. (2018)[3] show that standard momentum methods do not provide performance gain over SGD in several simple problem instances, and that their empirical benefits noticed in practice often arise only in large-batch regimes. This result is aligned with our findings that sophisticated optimizers are increasingly beneficial with larger batch sizes, and that the performance gap is significantly smaller for small batch sizes. While Kidambi et al. (2018) focus on vision experiments, we focus on the LLM optimization setting that is not characterized by multiple epoch training like
> vision and is often constrained to a limited token budget, stopping far before convergence and potentially leading to different training dynamics and scaling laws.
>
> Similarly to our work, Porian et al.(2025)[4] find that increasing the value of $\beta_2$ is critical to achieving a competitive performance with Adam at small batches. However, the way they choose $\beta_2$ at smaller batch sizes is fully based on a grid-search while we provide a scaling law for $\beta_2$ with batch size. We validate in our results throughout the paper that this new scaling law can be used in a zero-shot manner and lead to significant improvement in performance at smaller batch sizes.
> In summary, while these works offer theoretical and complementary perspectives, our contributions lie in challenging and putting to the test theoretical insights and empirical common wisdom specifically in the context of large language models optimization, providing recommendations that generalize across model size, datasets and optimizers, and deriving empirically-validated scaling laws that make our recommendations actionable. We acknowledge however that it is important to discuss these works, and we cite them indeed in the revised manuscript. We also discuss additional related work mentioned by reviewer 4.
>
> **Learning rate schedule:** We used a linear warmup for the first 5% of the training steps (from zero to peak learning rate), followed by cosine decay (from peak learning rate to zero). We use the same scheduler for all batch sizes. We updated our manuscript to reflect these experimental details.
>
> **Hardware utilization:** Indeed, we are referring to HBM bandwidth of the GPU/TPU. We store the model, optimizer, and full dataset on the GPU (the dataset is small), so there's no meaningful data movement between the GPU memory and CPU memory. Because modern GPUs/TPUs typically require an arithmetic intensity of several hundred FLOPs / byte to saturate tensor cores, they need to perform at least several hundred GEMM FLOPs per one byte of data loaded from HBM. In practice this means that every model parameter loaded from HBM needs to be used for at least several hundred (or even a few thousand) tokens in a batch. We would expect that increasing the batch size beyond this point provides no additional benefits in terms of hardware utilization, even if the accelerator has enough HBM to work with a larger batch size. Hence we recommend using the smallest batch size that achieves optimal or near-optimal hardware utilization.
>
> To give a clear idea about the wall-clock time in our setting, we report the training time for GPT-2 (124M) using our codebase with a TPU v6e-1 accelerator, SGD optimizer, and sequence length 1024 in the table below. We obtained the highest throughput when using batch size 4 (i.e. 4096 tokens per batch). Batch size 1 resulted in low arithmetic intensity and thus poor throughput. Conversely, batch size 32 required using gradient checkpointing, which increased the FLOPs per step and slowed down training. When training on a single accelerator, small batch sizes can result in good hardware utilization, but the minimum batch size required to achieve good hardware utilization will depend on both hardware and implementation details. Therefore, our general recommendation is to use the smallest batch size that results in good hardware utilization (e.g. in this specific example we would recommend using batch size 4).
>
> |                    |    BS=1        |  BS=4        |  BS=16      | BS=32       |
> | ---             | ---             | ---            |  ---           |     ---        |
> | Time         |  3h:32min  |    ⭐️2h:43min |   3h:14min  |   4h:16min |
>
> We updated our manuscript to clarify this point.
>
> Thank you again for your thoughtful feedback. We believe the extended related work discussion, clarifications and additional experiments meaningfully improve our paper, and we hope you will consider raising your score. Please let us know if you have additional questions we can address.

---

### Official Review · Reviewer_yNXQ · 2025-07-03

**Clarity:** 3
**Significance:** 3
**Originality:** 3
**Rating:** 5
**Confidence:** 5

**Summary:**

This paper investigates the training performance of LLM in small batch sizes.

**Questions:**

The paper cites in [40] that a "surge" phenomenon has been reported in the Adam optimizer. Could this issue potentially undermine the validity of the conclusions presented in this work?

**Ethical Concerns:**

["NO or VERY MINOR ethics concerns only"]

**Final Justification:**

The authors' newly presented experimental results and evidence have substantially alleviated my concerns regarding generalizability.

**Limitations:**

See Weakness.

**Paper Formatting Concerns:**

The manuscript explicitly references supplementary appendices in multiple sections but fails to include them in the submission.

Key instances include:

​​1. Section 4.1​​: "We provide additional experimental details in Appendix A."

​​2. Section 4.1​​: "We illustrate this effect in Appendix B.1 through a toy example."

**Quality:**

3

**Strengths And Weaknesses:**

**Strengths**

Some of the points raised in the paper are inconsistent with existing knowledge and offer new perspectives. For example, SGD can achieve performance similar to AdamW, and small batch training is more robust.

**Weaknesses**

1. The model used in the experiment is too small. The authors drew many of their conclusions based on a 42M model, and although these conclusions were verified using a 1.3B GPT-3 model, they were not verified using larger models. This raises concerns about the generalizability of these conclusions.
2. While Figure 1(b) compares the robustness to other hyper-parameters at batch sizes of 1 and 256, the continuous behavioral changes from small to large batch sizes are not thoroughly discussed across the entire range.
3. The paper appears structurally incomplete, and crucially, key appendixes are missing.

---

> ### Author Rebuttal · Authors · 2025-07-31
>
> Thank you for your feedback. We address your questions and concerns below.
>
> **Additional experiments at a larger scale:** In the submitted version of our paper, we pretrained models ranging from 30M active parameters to 1.3B (corresponding to $10^{17}$ – $10^{20}$ FLOPs). We trained all models (except the largest 1.3B model) to be Chinchilla optimal. We trained the 1.3B model for 10B tokens following the GPT-2 paper as well as the modded-nanoGPT repository. Since then, we have expanded our results in two ways: we tested more optimizers at the 124M and 1.3B scale and we tuned the learning rate of each optimizer at the 124M scale. And perhaps even more importantly, we added a fine-tuning experiment with a Gemma 3 4B, which was (over)trained on 4T tokens. We show that even this "late" in the training stage of a model, batch size 1 performs just as well as batch 16, even when fine-tuning for multiple epochs. We report these results in more detail below and in the main response.
>
> **Finetuning on the MATH dataset:** We fine-tune a non-instruction-tuned Gemma 3 (4B) model on the MATH dataset using various optimizers and batch sizes, allowing us to extend our experiments to a larger model size, a reasoning task, and a different regime where we train for multiple epochs. The evaluation metric is the percentage of problems correctly solved by the model. We compare full-parameter fine-tuning to Low-Rank Adaptation (LoRA), which freezes the pretrained weights and only updates small trainable low-rank matrices. LoRA is commonly used for fine-tuning because it reduces memory and compute costs while maintaining reasonable performance.
>
> However, our results reported in the table below show that we can instead address the memory constraints using a small batch size, and do full fine-tuning. Specifically, Adafactor (which has the same memory footprint as LoRA) and Adam with batch size one both outperform Adam with a larger batch size and LoRA with batch size one. This result suggests that constraining updates to a low-rank subspace, as in LoRA, is not necessary and may in fact deteriorate the performance. Full fine-tuning with small batches avoids gradient accumulation, favors memory-efficient optimizers, and achieves better results.
>
> | Adam, BS=16 | Adam, BS=1 | Adafactor, BS=1 | LoRA, BS=1 |
> | ---      | ---                  | ---                       |  ---                |
> | 18.2 ± 0.4%    | 18.7 ± 0.4%   | 18.7 ± 0.2%        | 16.7 ± 0.4% |
>
>
> **Grid-search for pretraining GPT-2 (124M):** We re-ran our grid-search experiments reported in Figure 1b using a larger GPT-2 (124M) model trained on 2.5B tokens of FineWeb, and we observe an even clearer trend of batch size 1 being more robust to hyperparameters. Here we show a binarized version of the plot where the dark regions correspond to validation loss <3.5 and the white regions correspond to validation loss >3.5. The smaller batch size is much more robust to both the momentum (y-axis) and the learning rate (x-axis) hyperparameters.
>
>              Batch size 1          Batch size 512
>           +----------------+     +----------------+
>     0.999 |████████████████|     |                |
>     0.99  |████████████████|     |                |
>     0.97  |████████████████|     |                |
>     0.9   |████████████████|     |     ███████    |
>     0.7   |████████████████|     |                |
>     0     |████████████████|     |                |
>           +----------------+     +----------------+
>           2⁻¹⁴ 2⁻¹² 2⁻¹⁰ 2⁻⁸     2⁻¹¹ 2⁻⁹  2⁻⁷  2⁻⁵
>             Learning rate          Learning rate
>
> We also tested Adafactor at 124M and 1.3B parameter scale for pretraining on FineWeb and in both cases it performed on par or even slightly better than Adam and AdamW with batch sizes 1 and 512 as shown in the table below.
>
> |               |  AdamW, BS=512 (baseline) |  Adam, BS=1, ${\beta_2}=0.95$ |  Adam, BS=1, ${\beta_2}=0.9999$ |    Adafactor, BS=1 |  SGD, BS=1 |
> | ---                    | ---                  | ---                       |  ---                | ---                    | ---                  |
> | GPT-2 (124M)    | 3.43           | 4.40                |  3.38               | 3.36⭐️                   | 3.52            |
> | GPT-3 (1.3B)      | 2.95               | 3.45              |  2.79            | 2.78⭐️              | 2.97                  |
>
>
> We would like to kindly note that running our grid-search experiments at the 1.3B scale would require around 2.7M TPU-v4-chip hours, at an approximate cost of $8.7M, which is not an accessible resource for us. However, several of our experiments with different models, datasets, and optimizers demonstrate the validity of our findings and recommendations at scale, including our fine-tuning experiments with the 4B-parameter Gemma 3 model.
>
> **Continuous behavioral changes from small to large batch sizes:** It is true that in Figure 1b, we only compare two batch sizes (1 and 256). However, we also measured the sensitivity of Adam to its hyperparameters in Figure 2. We tested batch sizes 1, 4, 16, 64, 256, 1024, and 4096, and saw a gradual increase in hyperparameter sensitivity with each increment of batch size.
>
> **Surge phenomenon:** The surge phenomenon described in [40] demonstrates that optimal learning rate for Adam increases with batch size until a critical point, beyond which it decreases and then rises again (thus, a “surge”). We believe this result is complementary to our findings and not in any conflict. For example, we observed in Figure 3 that the optimal learning rate for batch size 4096 was lower than for batch size 1024.
>
> **Appendix:** We included our Appendix in the supplementary materials, together with our source code. You can download it by clicking on the “Supplementary Material:  zip” button in our submission’s page on Openreview.
>
> Thank you again for your thoughtful feedback. We believe our revisions and additional experiments meaningfully improve our paper, and we hope you will consider raising your score. Please let us know if you have additional questions we can address.

---

> > ### Comment · Reviewer_yNXQ · 2025-08-04
> >
> > Thanks authors for the rebuttal, they have thoroughly addressed all my concerns. I will raise my score accordingly.

---

### Official Review · Reviewer_3pDd · 2025-07-03

**Clarity:** 2
**Significance:** 3
**Originality:** 2
**Rating:** 3
**Confidence:** 3

**Summary:**

This paper revisits small-batch training for language models, targeting the prevailing belief that large batches are essential for stability and efficiency. The authors show that, with proper hyperparameter tuning, in particular adjusting the second-moment decay in Adam based on token half-life, small batch sizes, down to batch size one, can achieve stable training, robustness to hyperparameter choices, and competitive or superior per-FLOP performance compared to larger batches. Notably, vanilla SGD without momentum performs on par with sophisticated optimizers in the small-batch regime, enabling significant memory savings. The paper argues against unnecessary gradient accumulation and advocates using the smallest batch size that maximizes hardware throughput.

**Questions:**

- You briefly mention Muon and Adafactor, but did you consider newer second-order or quasi-second-order optimizers (e.g., Sophia, Shampoo)? How might these perform in the small-batch regime relative to vanilla SGD?
- While small batches reduce memory and improve robustness, have you quantified the impact on wall-clock training time, given the reduced hardware throughput on modern accelerators? How should practitioners balance these trade-offs in practice?
- Did you evaluate whether models trained with small batches differ in downstream task performance (e.g., question answering, summarization) relative to large-batch-trained models?
- While small batches reduce memory and improve robustness, have you quantified the impact on wall-clock training time, given the reduced hardware throughput on modern accelerators? How should practitioners balance these trade-offs in practice?

**Ethical Concerns:**

["NO or VERY MINOR ethics concerns only"]

**Final Justification:**

The author did a very good effort in the rebuttal to answer all questions (thanks for that). However, and despite answer many questions, **I think two main fundamental issues still stand out (and both do not give confidence in the robustness of the proposed favorable effects of small batch sizes):**

- The paper largely provides empirical observations without rigorous theoretical explanations for why small-batch regimes enable competitive performance, which could limit the depth of understanding and generalization. The authors argue that experimentation alone is still a good indicator for effectiveness, however it is widely known that models can be sensitive to particular configurations/hyperparameters, in addition that not knowing why small batch sizes work also makes it unclear if this is a general effect, or just an artifact of the particular experimentation setup used.

- No exploring of the setting at which models are pretrained with different batch sizes and then evaluated using a fixed downstream fine-tuning setup. Without that, this is lack of concrete evidence that small batch size does indeed have the desired effect (after controlling for everything else).

**I respect the effort the authors put in the paper, and the rebuttal, yet I will maintain my score.**

**Limitations:**

Scale of large-model experiments, limited domain diversity, hardware efficiency trade-offs, absence of theoretical guarantees, incomplete optimizer coverage, and potential risks of underfitting.

**Quality:**

2

**Strengths And Weaknesses:**

Strengths:
- Challenges widespread assumptions about the necessity of large batch sizes in LM training, contributing valuable insights to optimization practice.
- Demonstrates that vanilla SGD without momentum can perform competitively, offering significant reductions in memory and compute requirements.

Weaknesses:
- The paper largely provides empirical observations without rigorous theoretical explanations for why small-batch regimes enable competitive performance, which could limit the depth of understanding and generalization (more on that on the two points below).
- While results extend to 1.3B-parameter models, the largest experiments remain less exhaustive (e.g., no full hyperparameter grid search), potentially leaving open questions about scalability to the largest frontier models.
- The primary experiments focus on FineWeb-Edu and related benchmarks; it is unclear whether findings generalize equally well to other domains or highly diverse corpora (e.g., code, multilingual data).
-  While advocating for small batch sizes, the paper briefly acknowledges throughput constraints but does not deeply quantify the performance penalties of very small batches on modern hardware accelerators.

---

> ### Author Rebuttal · Authors · 2025-07-31
>
> Thank you for your detailed review. We address your feedback below.
>
> **Novel results on other optimizers, datasets, and models:** Inspired by your feedback, we extend our experiments to the mC4 dataset, a multilingual variant of the C4 dataset comprised of natural text in 101 languages [1]. We also run additional fine-tuning experiments where we fine-tune and evaluate a non-instruction-tuned Gemma 3 (4B) [2] on the MATH dataset [3]. Finally, we test additional optimizers at scale. We describe our results below.
>
> **Scalability of our findings:** To demonstrate that our findings and recommendations extend to larger scale, we re-ran our grid-search experiments reported in Figure 1b using a larger GPT-2 (124M) model trained on 2.5B tokens of FineWeb, and we observe an even clearer trend of batch size 1 being more robust to hyperparameters. Here we show a binarized version of the plot where the dark regions correspond to validation loss <3.5 and the white regions correspond to validation loss >3.5. The smaller batch size is much more robust to both the momentum (y-axis) and the learning rate (x-axis) hyperparameters.
>
>              Batch size 1          Batch size 512
>           +----------------+     +----------------+
>     0.999 |████████████████|     |                |
>     0.99  |████████████████|     |                |
>     0.97  |████████████████|     |                |
>     0.9   |████████████████|     |     ███████    |
>     0.7   |████████████████|     |                |
>     0     |████████████████|     |                |
>           +----------------+     +----------------+
>           2⁻¹⁴ 2⁻¹² 2⁻¹⁰ 2⁻⁸     2⁻¹¹ 2⁻⁹  2⁻⁷  2⁻⁵
>             Learning rate          Learning rate
>
> We also tested Adafactor at 124M and 1.3B parameter scale for pretraining on FineWeb and in both cases it performed on par or even slightly better than Adam and AdamW with batch sizes 1 and 512 as shown in the table below.
>
> |               |  AdamW, BS=512 (baseline) |  Adam, BS=1, ${\beta_2}=0.95$ |  Adam, BS=1, ${\beta_2}=0.9999$ |    Adafactor, BS=1 |  SGD, BS=1 |
> | ---                    | ---                  | ---                       |  ---                | ---                    | ---                  |
> | GPT-2 (124M)    | 3.43           | 4.40                |  3.38               | 3.36⭐️                   | 3.52            |
> | GPT-3 (1.3B)      | 2.95               | 3.45              |  2.79            | 2.78⭐️              | 2.97                  |
>
> We would like to kindly note that running our grid-search experiments at the 1.3B scale would require around 2.7M TPU-v4-chip hours, at an approximate cost of $8.7M, which is not an accessible resource for us. However, several of our experiments with different models, datasets, and optimizers demonstrate the validity of our findings and recommendations at scale, including our fine-tuning experiments with the 4B-parameter Gemma 3 model.
>
> **Pretraining on the mC4 dataset:** Inspired by your recommendation to extend our experiments to a diverse multilingual dataset, we used Adafactor with batch size 1 and AdamW with batch size 512 to train GPT-2 (124M) for 2.5B tokens on the m4C multilingual dataset tokenized using the T5 tokenizer. We used the same hyperparameter values as in the table above (where the learning rate was tuned for each optimizer) and obtained a validation loss of **0.96** for Adafactor with batch size 1 and **0.99** for AdamW with batch size 512. We specifically chose Adafactor for this comparison since it is a simple optimizer with an extremely small memory footprint. We find it interesting that when a small batch size is used, such a simple optimizer performs competitively with the gold-standard AdamW, even in the multilingual setting.
>
> **Finetuning on the MATH dataset:** We fine-tune a non-instruction-tuned Gemma 3 (4B) model on the MATH dataset using various optimizers and batch sizes, allowing us to extend our experiments to a larger model size, a reasoning task, and a different regime where we train for multiple epochs. The evaluation metric is the percentage of problems correctly solved by the model. We compare full-parameter fine-tuning to Low-Rank Adaptation (LoRA), which freezes the pretrained weights and only updates small trainable low-rank matrices. LoRA is commonly used for fine-tuning because it reduces memory costs.
>
> However, our results reported in the table below show that we can instead address the memory constraints using a small batch size, and do full fine-tuning. Specifically, Adafactor (which has the same memory footprint as LoRA) and Adam with batch size one both outperform Adam with a larger batch size and LoRA with batch size one. This result suggests that constraining updates to a low-rank subspace, as in LoRA, is not necessary and may in fact deteriorate the performance. Full fine-tuning with small batches avoids gradient accumulation, favors memory-efficient optimizers, and achieves better results.
>
> | Adam, BS=16 | Adam, BS=1 | Adafactor, BS=1 | LoRA, BS=1 |
> | ---                    | ---                  | ---                       |  ---                |
> | 18.2 ± 0.4%    | 18.7 ± 0.4%   | 18.7 ± 0.2%        | 16.7 ± 0.4% |
>
> **Rigor and generalizability of our work:** While our work is primarily empirical rather than theoretical, we follow a rigorous scientific approach. We clearly specify our hypotheses about small batch size training, carefully remove confounding factors such as weight decay and variable token budget, and test one hyperparameter at a time thanks to our grid-search. This rigorous approach enables us to derive insights and formulate theories that are testable and falsifiable. Additionally, our experiments span multiple model scales, optimizers, and datasets. With the newly added fine-tuning experiments, we extend our analysis to both single-epoch and multiple-epoch training regimes, therefore broadening the scope and generalizability of our findings. To provide intuition behind our findings, we included in Appendix B (added to the supplementary material) a toy problem illustrating why momentum is necessary for stable training with large batch sizes but not required when training with small batches.
>
> **Throughput and trade-offs:** Our general recommendation is to use the _smallest_ batch size that results in good hardware utilization. To saturate their matrix multiplication units, modern ML accelerators typically require an arithmetic intensity of several hundred FLOPs / byte, i.e., they need to perform at least several hundred GEMM FLOPs per one byte of data loaded from global memory. In our case, this means that every model parameter loaded from global memory needs to be used for at least several hundred tokens in a batch.
>
> **Reporting wall-clock time:** We now report wall-clock training times for different batch sizes using our codebase on a TPU v6e-1 accelerator. We find that very small batches have low arithmetic intensity, while very large batches incur overhead from gradient checkpointing. In our setup, BS=4 achieves the best throughput. To re-iterate, we recommend using the smallest batch size that achieves good hardware utilization, and this threshold varies across hardware platforms.
>
> |                    |    BS=1        |  BS=4        |  BS=16      | BS=32       |
> | ---             | ---             | ---            |  ---           |     ---        |
> | Time         |  3h:32min  |    ⭐️2h:43min |   3h:14min  |   4h:16min |
>
>
> **Considering newer second-order or quasi-second-order optimizers:** Our choice to use Muon and not other second-order or quasi-second-order optimizers is motivated by the fact that Muon (2024) is the newest and most performant optimizer for large language model training. Muon has been used in several recent state-of-the-art settings, including the NanoGPT speedrun, CIFAR-10 speedrun, and training the Kimi K2 models. In contrast, Shampoo (2018) has significant memory overhead, and Sophia (2023), while promising, underperforms compared to Muon for LLM training. Therefore, we believe that Muon is the most relevant quasi-second-order method for our work. Our choice of Adafactor as another baseline stems from its extremely low memory footprint.
>
> **Impact on downstream performance:** Our additional fine-tuning experiments show that small-batch training performs competitively in downstream tasks under memory constraints. However, we did not explore the setting where models are pretrained with different batch sizes and then evaluated using a fixed downstream fine-tuning setup. We agree this is an interesting direction, and consider it a promising avenue for future work.
>
> Thank you again for your detailed feedback, we believe it has positively impacted our paper and extended the scope of our experiments. We made a significant effort to address your points and hope that you would consider raising your score. Please let us know if you have any other questions that we can answer.
> ____
>
> References:
>
> - [1] Xue, L., Constant, N., Roberts, A., Kale, M., Al-Rfou, R., Siddhant, A., Barua, A. and Raffel, C., 2020. mT5: A massively multilingual pre-trained text-to-text transformer. arXiv preprint arXiv:2010.11934.
> - [2] Team, G., Kamath, A., Ferret, J., Pathak, S., Vieillard, N., Merhej, R., Perrin, S., Matejovicova, T., Ramé, A., Rivière, M. and Rouillard, L., 2025. Gemma 3 technical report. arXiv preprint arXiv:2503.19786.
> - [3] Hendrycks, D., Burns, C., Kadavath, S., Arora, A., Basart, S., Tang, E., Song, D. and Steinhardt, J., 2021. Measuring mathematical problem solving with the math dataset. arXiv preprint arXiv:2103.03874.

---

### Note · Authors · 2025-08-15

We thank the reviewers for their feedback, which helped us substantially improve our manuscript. Based on the feedback, we extended our experiments to larger models, tested a multilingual dataset, added fine-tuning, and expanded the discussion of related work, as detailed in our discussion with the reviewers.

While our work builds on observations documented in prior works, this paper introduces multiple novel and practical findings. In particular, we demonstrate that smaller batch sizes are more robust to hyperparameter values and optimizer design, and we show how this enables full fine-tuning in a memory-constrained setting. We also introduce a novel scaling heuristic for $\beta_2$ that generalizes across model scales and we apply it to correct the results from Xiao (2024). These findings offer actionable insights to make the training of language models more efficient and accessible.

---

### Decision · Program_Chairs · 2025-09-17

**Decision:**

Accept (poster)

**Comment:**

The paper presents a thorough empirical study challenging the prevailing belief that large batch sizes are essential for stable and efficient language model training. The authors demonstrate that, with appropriate hyperparameter scaling—particularly adjusting the second-moment decay parameter β₂ based on token half-life—small batch sizes, even down to one, enable stable training, increased robustness to hyperparameters, and competitive or superior per-FLOP performance compared to larger batches. Notably, they show that vanilla SGD without momentum performs on par with more sophisticated optimizers, leading to significant memory savings and practical benefits in memory-constrained fine-tuning scenarios.

The strengths of this work lie in its extensive experimental design, inclusion of diverse optimizers (Adam, Adafactor, Muon), evaluation on multiple datasets (FineWeb, mC4, MATH), and scaling experiments up to 1.3B-parameter models.

However, the key concern is that the findings are primarily empirical and lack a rigorous theoretical framework explaining why small-batch regimes generalize effectively, limiting broader understanding and transferability.

During the rebuttal, the authors addressed many reviewer concerns by expanding experiments to larger models, multilingual data, and fine-tuning setups, demonstrating consistent trends favoring small batches and showing that Adafactor can achieve full fine-tuning performance with minimal memory overhead.

Despite the lack of deeper theoretical insights, the comprehensive experimentation, novel scaling heuristics, and practical relevance make this a valuable and timely contribution. I recommend acceptance given its strong empirical findings, actionable recommendations for practitioners, and meaningful impact on optimization strategies for language model training.